# A Comprehensive Approach for Detecting Brake Pad Defects Using Histogram and Wavelet Features with Nested Dichotomy Family Classifiers

**DOI:** 10.3390/s23229093

**Published:** 2023-11-10

**Authors:** Sakthivel Gnanasekaran, Lakshmi Pathi Jakkamputi, Jegadeeshwaran Rakkiyannan, Mohanraj Thangamuthu, Yogesh Bhalerao

**Affiliations:** 1School of Mechanical Engineering, Vellore Institute of Technology, Chennai 600127, India; sakthivel.g@vit.ac.in; 2Centre for Automation, Vellore Institute of Technology, Chennai 600127, India; lakshmipati@vit.ac.in; 3Department of Mechanical Engineering, Amrita School of Engineering, Amrita Vishwa Vidyapeetham, Coimbatore 641112, India; 4Department of Mechanical Engineering and Design, School of Engineering, University of East Anglia, Norwich Research Park, Norwich NR4 7TIJ, UK; y.bhalerao@uea.ac.uk

**Keywords:** histogram feature, wavelet feature, nested dichotomy, class-balanced nested dichotomy, data-near-balanced nested dichotomy

## Abstract

The brake system requires careful attention for continuous monitoring as a vital module. This study specifically focuses on monitoring the hydraulic brake system using vibration signals through experimentation. Vibration signals from the brake pad assembly of commercial vehicles were captured under both good and defective conditions. Relevant histograms and wavelet features were extracted from these signals. The selected features were then categorized using Nested dichotomy family classifiers. The accuracy of all the algorithms during categorization was evaluated. Among the algorithms tested, the class-balanced nested dichotomy algorithm with a wavelet filter achieved a maximum accuracy of 99.45%. This indicates a highly effective method for accurately categorizing the brake system based on vibration signals. By implementing such a monitoring system, the reliability of the hydraulic brake system can be ensured, which is crucial for the safe and efficient operation of commercial vehicles in the market.

## 1. Introduction

The hydraulic brake system is a critical component of an automobile and is responsible for controlling the vehicle’s speed and reducing the stopping distance. Any malfunction in the brake system can have a detrimental impact on the vehicle’s stability and the safety of its passengers. According to the National Motor Vehicle Causation Survey, it has been revealed that 22% of accidents occur as a result of brake system failures [1]. In 2007, BMW recalled over 155,000 Sports Utility Vehicles (SUVs) due to brake fluid issues. This recall was initiated to address concerns regarding the performance and functionality of the brake system in these vehicles. The recall aimed to ensure the safety of the vehicles and their occupants by rectifying the brake-fluid-related issues. Similarly, Chrysler also faced a recall in 2007 involving approximately 60,000 vehicles. The recall was due to braking issues, specifically, a lack of success in the braking system. Chrysler undertook this recall to address the identified deficiencies and ensure the proper functioning of the brakes in the affected vehicles [2].

These recalls demonstrate the commitment of automotive manufacturers to prioritize safety and take prompt action to rectify any issues related to braking systems. This statistic underscores the significance of maintaining a reliable and properly functioning brake system to prevent accidents and ensure the safety of both drivers and passengers. Regular maintenance, inspections, and testing are crucial for vehicle owners, manufacturers, and regulatory authorities to mitigate the risk of brake system failures. By implementing effective monitoring systems and taking preventive measures, potential issues can be identified early on, allowing for timely corrective actions and reducing the occurrence of accidents stemming from brake system failures. Examining such systems for defects will decrease the total crashes and enhance safety to a certain extent. Currently, a specific method for monitoring brake system failure is unavailable. Therefore, it is essential to develop a suitable method to monitor failures related to the brake system. The brake system may become flawed owing to pad wear, mechanical fade, inadequate oil pressure in the control cylinder, and oil leakage. If these flaws are not observed, accidents will take place. Hence, proactive maintenance is recommended to prevent accidents and can be attained through continuous monitoring [3,4].

Condition monitoring (CM) is continuously observing a system’s operational attributes to initiate necessary steps before the occurrence of a breakdown. Many CM methods utilize devoted sensors and data assessment tools to examine a specific type of disparity in functional attributes [5]. Thomas et al. outlined a new communications module for identifying brake faults through sensors. The module detects the malfunction through the recorded signals from the brake system [6]. A method was patented for monitoring the vehicle condition and generating a maintenance plan according to the vehicle condition [7,8,9]. A new approach for monitoring the aircraft brake was proposed to determine the brake condition [10].

Nevertheless, to the best knowledge of the authors, there is no available solution for monitoring the brake pad conditions. Hence, a new method is required to diagnose brake pad-related failure. Fault diagnosis is a sub-area of CM in which specific system conditions are continuously studied. Fault identification can be performed in numerous ways, like visual testing, acoustic emission (AE) [11], thermal imagers, ultrasonic, and vibration signals [12]. Visual testing is the most commonly used technique in the industry. However, visual inspection has drawbacks since it requires skilled laborers who can identify the problems through experience. Hence, it is unsuitable for monitoring hidden components such as the brake system [13].

A passive infrared thermal imager was utilized to monitor the weld quality of the specimen, but sophisticated equipment was needed to measure the temperature variance [14]. Since it is an offline process, it is also unsuitable for brake fault diagnosis. AE can be used to monitor the machining process [15]. The bearing fault in a helicopter gearbox was identified by measuring the AE and vibration signals [16]. The spur gear condition was monitored using AE and vibration signals [17]. The study proved that vibration analysis is more appropriate for fault than AE signal analysis. However, the AE signal acquisition is complicated in machine components such as a brake.

Recently, most of the fault diagnostic studies focused on the vibration signal. The vibration signals are examined through frequency domain analysis, wavelet analysis, waveform examination, spectral analysis, statistical learning, histogram learning, etc. These examinations gave the data needed to decide on the maintenance plan. The consequences of such examinations are utilized for fault investigation to determine the fault’s original cause. The brake system produces vibrations under various working circumstances. The measured vibration signals correlate precisely with the fault conditions, which are used to identify the defects in the system. Therefore, vibration signals can be considered appropriate for condition monitoring through feature-based analysis.

The feature is a specific, measurable characteristic of an anomaly being examined. Selecting data, separating, and categorizing independent features is essential in classification, pattern recognition, and regression. Numerous features, specifically statistical [18], histogram [19], and wavelet [20], can be obtained from the acquired signals. The application of a wavelet plays a crucial role in CM. The main advantage of the wavelet transform (WT) is to denoise or compress a signal without reducing the original signal and computational time. WT could be successfully applied to the different fault diagnosis studies. Hence, WT has been used for diagnosing brake failure in this study.

After extracting the features, the most significant features have to be selected. The attribute evaluator, the influence of the number of features, and Decision Tree (DT) are the three most important methods for selecting the features. DT utilizes a graphical structure to select the most significant features [21]. The attribute evaluator calculates the value of a characteristic subset by studying each feature’s analytical capacity and severance level. The influence of the number of features is one of the most helpful techniques in feature selection. Instead of taking all the features, the performance of the specific features is tested using this examination. The influence of several features study was applied to diagnose the brake faults and identify the best feature combination that produces the maximum classification accuracy [22].

Feature classification is the last phase in the classifier. The selected features should be classified for predicting performance. The commonly used classifiers are Decision Tree [23,24], Fuzzy [25], Artificial Neural Network [20], Bayes Net [26], Naive Bayes [20], Support Vector Machine (SVM) [27], Convolutional Neural Network (CNN) [28] and the Hidden Markov Model [29]. Different classifiers monitored the tool wear using vibration signals with the discrete wavelet feature [20]. So, there is considerable scope for the wavelet features with other machine learning algorithms in the brake monitoring system. The nested dichotomy (ND) classifier was used for fault diagnosis in the hydraulic brake system. They proved that the class-balanced nested dichotomy (CBND) prediction accuracy was more than that of the SVM and DT classifiers. The authors used the statistical feature alone for classification. The other features can also be considered to make a more detailed study.

To the authors’ best knowledge, there is no literature on CM of a brake system using histogram and wavelet features. Hence, histogram and wavelet features are considered to monitor the hydraulic brake system. An attribute evaluator is used for feature selection. The feature selection through the attribute evaluator is validated using the influence of the number of features studied. The preferred wavelet and histogram features were categorized using the family of ND classifiers, and the results were compared and discussed.

## 2. Experimental Procedure

Experimentation was conducted to supervise the brake system using histogram and wavelet features extracted from the vibration signals. A brake system test rig was used to perform the experiments. The experimental test rig and procedure are briefed in the subsequent sections.

### 2.1. Experimental Test Rig

A commercial vehicle, as depicted in Figure 1, was utilized for the experiment. Its drive shaft consistently operated at 331 rpm on a test bed equipped with rollers. A piezo-electric uni-axial accelerometer (Make: Dytran, Sensitivity: 10 mV/g) was securely positioned at the center of the flat surface of the brake caliper, as illustrated in Figure 1c. As the drive shaft rotated, various dynamic forces and loads impacted the brake pad and caliper assembly. These forces resulted from the interaction between the brake pad and the brake rotor, leading to vibrations within the brake pad and caliper assembly. As the brake pad wears, it undergoes changes in both rigidity and mass distribution, which, in turn, cause fluctuations in the assembly’s stiffness and mass. These variations in vibration within the brake pad assembly, associated with the brake pad’s stiffness, are crucial parameters for understanding and identifying brake pad defects. The attached piezo-electric accelerometer captures these variations in vibration signals, which are then processed using the NI 9234 data acquisition (DAQ) system, converting them into digital data. The encoded data from the DAQ system is subsequently processed and presented in the LabVIEW 2021 software, providing results in the time domain as presented in Figure 1b. The experiments were performed with different brake pad conditions, as depicted in Figure 1d.

### 2.2. Experimental Procedure

The frequently occurring failures, such as Brake Oil Spill (BOS), Drum Brake Mechanical Fade (DBM), Disc Pad Wear (Uneven) Inner (DBP(UE)I), Disc Pad Wear (Uneven) Inner and Outer (DBP(UE)IO), Disc Pad Wear (Even) Inner (DBPI), Disc Pad Wear (Even) Inner and Outer (DBPIO), Drum Brake Pad Wear (DBPW), Air in Brake Fluid (ABF), and Reservoir Leak (RL) were considered for simulating the fault conditions. The vibration signals were acquired from the test setup with good and various simulated faulty brake conditions. The LabVIEW program used for vibration signal acquisition is shown in Figure 2. The data acquisition was performed with the following specifications:
➢Sample length and no. of samples: 2^13^ and 55 (arbitrarily chosen);➢Sampling frequency: 24 kHz (as per the Nyquist sampling theorem);➢Wheel speed and brake force: 331 rpm (~30 KMPH) and 68.7 N.

The wheel speed, which was 331 rpm, was measured and controlled using a tachometer. The brakes were applied gradually and evenly to prevent RPM fluctuations, and, importantly, as the brakes were applied, minor manual adjustments were made to the throttle to ensure a consistent RPM was maintained.

**Figure 2 sensors-23-09093-f002:**
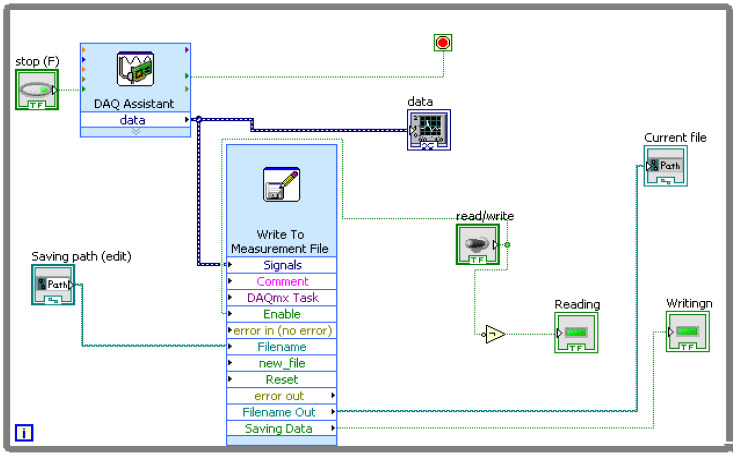
LabVIEW program for data acquisition.

## 3. Histogram Features

The histogram data is used as a feature in the brake CM. The magnitude range of the vibration is split into various sub-ranges (bins) from the smaller to the larger value of the vibration signal. The quantity of information focuses on whose magnitude of vibration value cataracts inside a specific bin are calculated. The bin and count frame the histogram’s x and y axes. The histogram for each defect is plotted utilizing vibration motions as an independent graph. The goal is to discover the bins whose *y*-axis values are the same for a specific defect but not the same for another defect. The histogram for a specific brake defect might be small; however, it might be huge for other defects. The bin range chosen should suit all defects. The strategy for estimating the base of least magnitude is as follows:
Compute the least magnitude of each signal in a specific condition;Repetition of step (i) for all conditions;Identify the least of those least magnitudes.

Similarly, the base of the largest magnitude is computed. The bin width should be fixed so that the bin height is diverse for various brake conditions. It requires not to be valid for all widths of the bin, but rather, at least a few of them should be taken after this measure is utilized as a feature for recognizing different conditions. This study used the bin width and range to plot the histogram, as shown in Figure 3a. The vibration value between the larger and smaller values was divided by a number of bins. The larger and smaller value range is divided into 69 bins (2 to 70). Figure 3a–j shows the histogram obtained from the vibration signals.

## 4. Wavelet Filter

The wavelet transform (WT) is a prevailing time and frequency domain signal tool. WT can be continuous or discrete, permitting the investigation of fleeting, irregular, or local parts. The continuous WT can expose more insights about a signal but is computationally intensive. For fault analysis, discrete wavelet transform (DWT) aims to characterize a signal proficiently with fewer parameters and computation time. DWT is an appropriate method for this study [30].

### 4.1. Wavelet Transform

The WT is drawn as
(1)$$\begin{array}{l} WT_{f}(p,q)=f(t).\psi_{p,q}(t)\\ WT_{f}(p,q)=\frac{1}{\sqrt{p}}\int_{0}^{\infty }f(t)\psi \left(\frac{t-q}{p}\right)dt \end{array}$$
where $\psi_{p,q}(t)$ is the alternative of the “Mother wavelet”, which is considered as
(2)$$\psi_{p,q}(t)=\frac{1}{\sqrt{p}}\int_{0}^{\infty }\psi \left(\frac{t-q}{p}\right)$$
where *t*, *p*, *q* ϵ *R*, and *p* ≠ 0 are continuously varying quantities, p is the scale factor, *q* is a shift factor, and $\frac{1}{\sqrt{p}}$ is used for energy conservation. If *p* is small, extreme-frequency components are analyzed, and vice versa.

### 4.2. Discrete Wavelet Transform

DW performs scaling and translation procedures in distinct phases, restraining the selection of scales (*a*) and translation (*τ*) to specific numbers. However, the examination is satisfactorily precise and articulated in the following equation:(3)$$\psi_{j,k}(t)=\frac{1}{\sqrt{s_{0}^{j}}}\psi \left(\frac{t-k\tau_{0}s_{0}^{j}}{s_{0}^{j}}\right)$$
where *j* and *k* are integers.

The time scale is tested at a distinct interval due to discretizing the wavelets, called decomposition, and succeeds a sequence of wavelet coefficients [31]. The wavelet features were obtained for every condition utilizing the Daubechies wavelets “*db1*” to “*db10*”. The wavelets studied for this examination are provided below:
Haar—‘haar’;Daubechies—‘db’;Symlet—‘sym’;Coiflets—‘coif’;Biorthogonal—‘bior’;Reverse biorthogonal—‘rbio’;Discrete approximation of Meyer—‘dmey’;Fejer–korovkin—‘fk’.

The WT removed the noise from the raw vibration signal and reconstructed the signals. The successive approximation filters certain more high-frequency data from the raw signal. Using this method, substantial data loss is possible. Therefore, a new method is required for denoising. Optimal denoising demands a more sensitive approach known as thresholding, which includes discarding a small number of progressive elements that exceed a specific threshold. Thresholding is reformed from level to level. For the current study, level 5 approximation was applied. The eight types of wavelet features were extracted for every condition. Figure 4a–j shows the denoised sample signal attained through DWT for the Coiflets wavelet. From the denoised signals, relevant statistical features were collected.

## 5. Feature Selection

The attribute evaluator, DT, and the influence of the number of features studied were used to select the most significant features. This process is called feature selection. The attribute evaluator was utilized to find the best feature set from the obtained features. In this study, the two different search methods, Correlation-based Feature Selection Subset Eval and Best First Search were applied to select the sequence of the significant features. The chosen feature was tested through the effect of the number of features studied. The first element suggested by the attribute evaluator was classified first, and the accuracy was recorded. Then, the second feature recommended by the evaluator was combined with the first feature and classified. The classification accuracy with two feature combinations was also recorded. The process was repeated until all the features were combined and classified. The best feature combination that only provides maximum accuracy was selected for further study. The DT was used for choosing the most significant features from the histogram feature.

## 6. Feature Classification

The selected features should be classified to obtain the classification result. A system of ND is employed to disintegrate a multinomial problem into multiple binary problems. The disintegration is described using a binary tree. A set of class labels is stored in each tree node, and a binary classifier from the training data. The training data is divided into two subsets conforming to the two meta-classes, and one subset of training data is considered a true example.

In contrast, the other subset of testing data is considered a false example. The two next-in-line root nodes receive two subsets of the true class labels with their conforming training data. A tree is constructed by continuously employing this process. Lastly, this process succeeds a leaf node if the node has only one label. The choice of tree structure significantly impacts the accuracy.

The CBND method depends on regulating the number of conditions at every node. As an alternative to examining each possible tree’s space, CBND examined the space of each balanced tree. The quantity of possible trees in CBND is lesser than the entire tree of ND. The detailed algorithm for developing a system of CBND is given in [32]. The usual difficulty with the CBND method is that few multinomial issues are not balanced and more populated than others. A CBND does not have additional data balanced. This can undesirably affect the computational time of the algorithm. DNBND arbitrarily assigns the classes to two subsets until the training data size in one subset surpasses 50% of training data at the node. With the DNBND algorithm, assortment suffers when class dispersal is disturbed. The number of classes depends on the class distribution in the data.

## 7. Results and Discussion

The wavelet and histogram features were extracted, and the most significant features were chosen. The chosen features were classified using ND family classifiers.

### 7.1. Histogram Feature Classification Using Nested Dichotomy Family Classifiers

The DT algorithm chose the most significant features [23]. All the 69 bins ranging from 2 to 70 were classified using the DT algorithm one after another. The classification accuracy for different bin ranges is shown in Figure 5. The ND classifier delivers a maximum classification accuracy for the 59th bin of 97.64%. The algorithm uses decision rules for classification. These decision rules are graphically represented as a DT, as shown in Figure 6. Hence, the graphical DT shows the contributing bin ranges. Even though 59 bin ranges were used as input in the algorithm, it uses only 7 for classification. Figure 6 selected the contributing bin range (59^th^) alone which have the maximum classification accuracy (97.64%). The features were categorized using various ND-family algorithms, including ND, CBND, and DNBND. The parameters used for the classification are given below:Random no. seed: 1;No. of trees: 10;Base learner: best first tree;Tree depth: unlimited.

**Figure 5 sensors-23-09093-f005:**
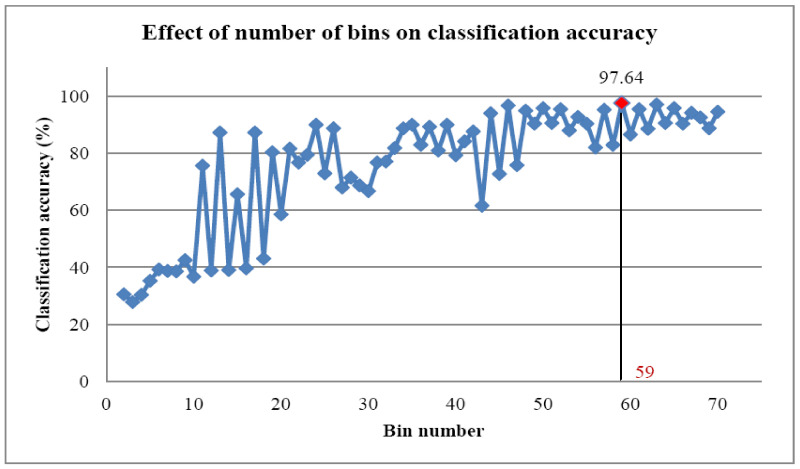
Classification accuracy vs. number of bins.

**Figure 6 sensors-23-09093-f006:**
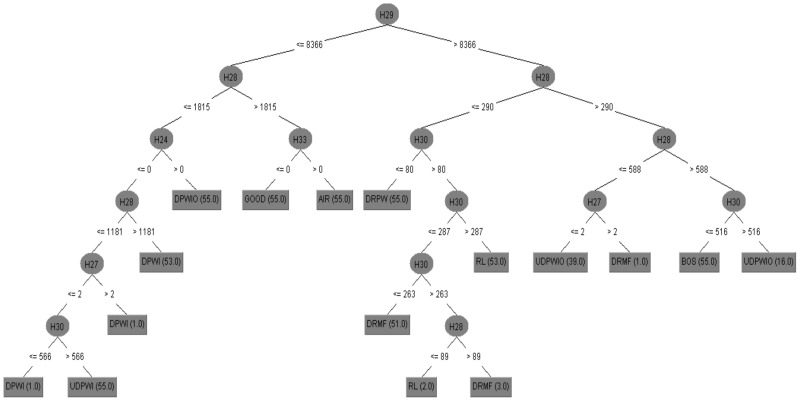
Graphical DT with histogram features.

The categorization accuracy is displayed in Figure 7. In Figure 7, the first cell signifies the number of data conforming to the “GOOD” class. The sum of data points in the first row is 55. All are correctly categorized and yield a classification accuracy of 100%. Likewise, the other element in the first row represents the false categorization. There are no data points on the first row beside the first column. This means that there is no false classification. Comparably, the sixth row sixth column signifies the DBPI condition. Among the 55 datapoints, 53 were categorized accurately, and two were wrongly classified under the DBP(UE)I condition. The classification accuracy was estimated as 96.4%. Among the 550 datapoints, 538 were accurately classified, with an overall classification accuracy of 97.5%.

The rows and columns represent the anticipated and actual conditions, respectively. The diagonal cells correspond to data points that are appropriately categorized. The off-diagonal cells represent wrongly categorized data. The no. of data is shown in each cell, and the exact prediction accuracy is shown in the last cell. The last two columns on the right show the % of all the data predicted to belong to each appropriate (precision) and wrongly classified (false discovery rate) condition. The last two rows at the bottom show the % of the data predicted appropriately (True Positive Rate (TPR)) and wrongly (False Negative Rate (FNR)) classified condition. The cell in the bottom right shows the overall accuracy.

Similarly, the selected features were classified using CBND and DNBND classifiers, and the corresponding classification accuracies were found. The overall accuracy for CBND and DNBND was 97% and 99%, respectively. Figure 8 and Figure 9 represent the confusion matrix obtained for CBND and DNBND with histogram features.

### 7.2. Wavelet Feature Classification Using ND Family Classifiers

Eight families of wavelets were selected, and a denoised signal was obtained from each family. From the denoised signal, twelve statistical features were extracted. The attribute evaluator was used to select the most significant features, and nine statistical features were acknowledged and considered as input for the classifier. The nominated nine features were appraised using the influence of the no. of feature study. Table 1 indicates the influence of the no. of features studied performed on the Coiflets wavelet. The CBND produces a better accuracy (99.45%) with the top eight features.

The feature assortment process was applied for all wavelet families, and the resultant accuracy was recorded and presented in Table 2. In Table 2, the Coiflets wavelet (Coif) produced a maximum classification accuracy compared to other wavelets. Therefore, the Coiflets wavelet was selected. From the influence of the no. of feature study, the eight features that contributed to the classification were chosen for the study. They were as follows: (1) Mean, (2) Standard Error, (3) Median, (4) Standard Deviation, (5) Kurtosis, (6) Skewness, (7) Range, and (8) Minimum. The classification accuracy for the ND family (ND, CBND, and DNBND) algorithms was found. Figure 10 represents the confusion matrix for ND. Figure 10 reveals the overall classification accuracy of 99%.

Comparably, the remaining data points in the first row signify the false classification details. All other values are zero in the first row other than the first element, revealing no false classification. Similarly, the last element in the last row represents RL. Among the 55 datapoints, 53 belong to RL and are accurately categorized, and 2were wrongly categorized as the DBM condition. Among the 550 datapoints, 543 were categorized accurately, with an accuracy of 99%.

Likewise, the classification accuracy was computed for CBND and DNBND. Both the classifier produced 99.45% and 99.27% as classification accuracy, respectively. Figure 11 and Figure 12 show the graphical representation of the confusion matrix attained for the CBND and DNBND. The overall accuracy of the ND classifier is presented in Table 3. The CBND gives an enhanced accuracy of 99.45% among the three ND family classifiers. Among the 550 datapoints, 547 were accurately classified. Denoising using wavelet significantly reduced noise and increased classification accuracy.

### 7.3. Comprehensive Accuracy by Condition

The specific condition implementation of the ND can be studied using comprehensive accuracy by condition study, termed as TPR, FPR, Precision, and ROC. The TPR corresponds to the accurately identified conditional class, and the FPR is the rate at which the classes are wrongly classified. For a machine learning algorithm, TPR must be 1, and FPR should be 0. This study’s TPR and FPR are closer to 1 and 0, respectively. Similarly, the other parameter values are also closer. Table 4 presents the classification accuracy for CBND for wavelet feature extraction.

### 7.4. Comparative Analysis

The captured raw vibration signals were denoised through WT. The carefully chosen features were categorized using ND, CBND, and DNBND, and the overall accuracy is shown in Table 5. The results of wavelet features were compared with histogram features. Table 5 shows the WT of the vibration signal with CBND, revealing a maximum overall accuracy of 99.45%. The comparative results show that the wavelet features and ND outperform the statistical features and all other classifiers considered in this study. Hence, the wavelets with CBND can be considered an appropriate classifier for diagnosing brake faults.

## 8. Conclusions

The paper discussed the wavelet and histogram features for supervising the brake system using vibration signals. The present study considered nine good simulated faults of the brake system. The vibration signals were recorded under all simulated defects. The wavelet and histogram features were extracted from the vibration signal, and the most significant features were selected. Then, the preferred features were classified using ND, CBND, and DNBND. The results reveal that the CBND classifier performed better than the ND and DNBND in classification accuracy. The competency of the wavelet features was evaluated with the histogram features, and the wavelet feature yielded an overall prediction accuracy of 99.45%. Henceforth, the CBND with wavelet features can be recommended for online brake fault diagnostics. This research is protracted with actual road conditions for monitoring the hydraulic brake system.

## Figures and Tables

**Figure 1 sensors-23-09093-f001:**
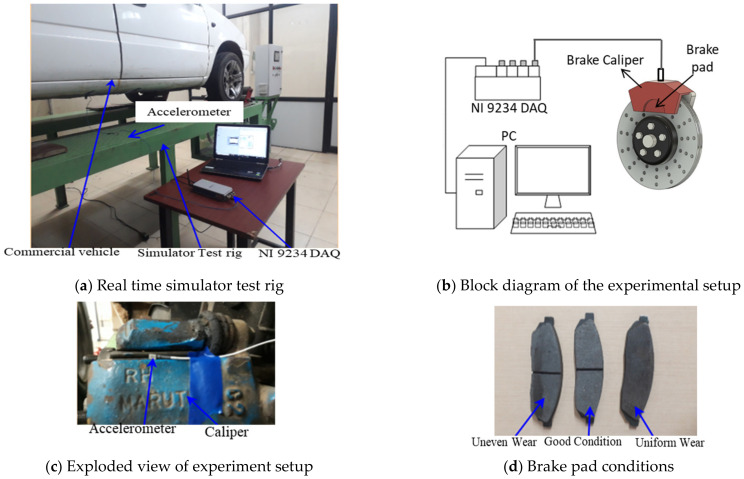
Experimental test rig.

**Figure 3 sensors-23-09093-f003:**
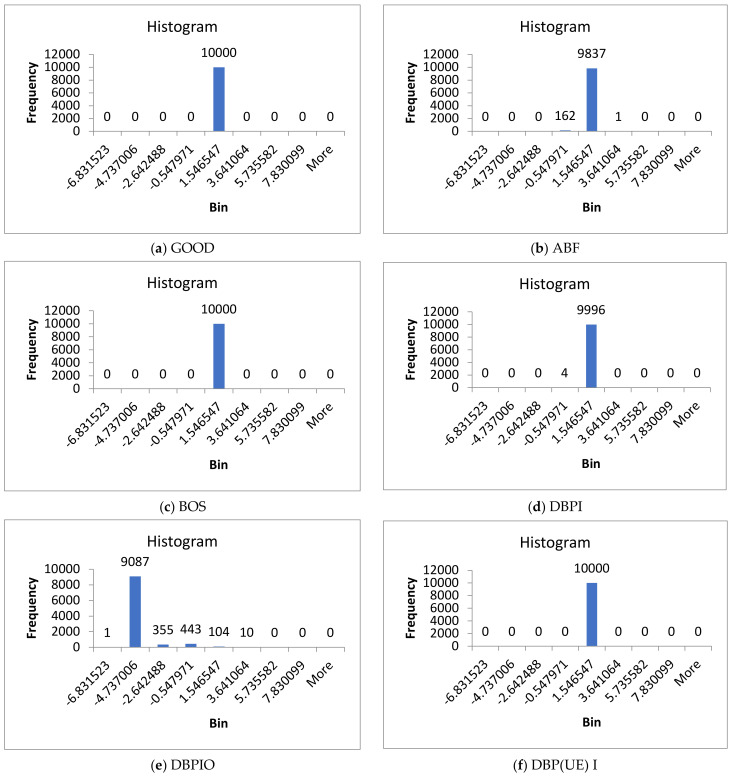
Histogram of different faulty conditions for 7 bin range.

**Figure 4 sensors-23-09093-f004:**
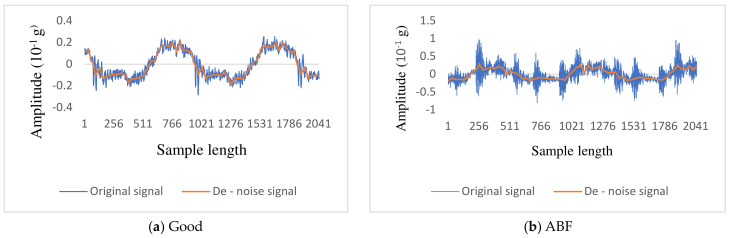
Original and denoised signals for various faulty conditions.

**Figure 7 sensors-23-09093-f007:**
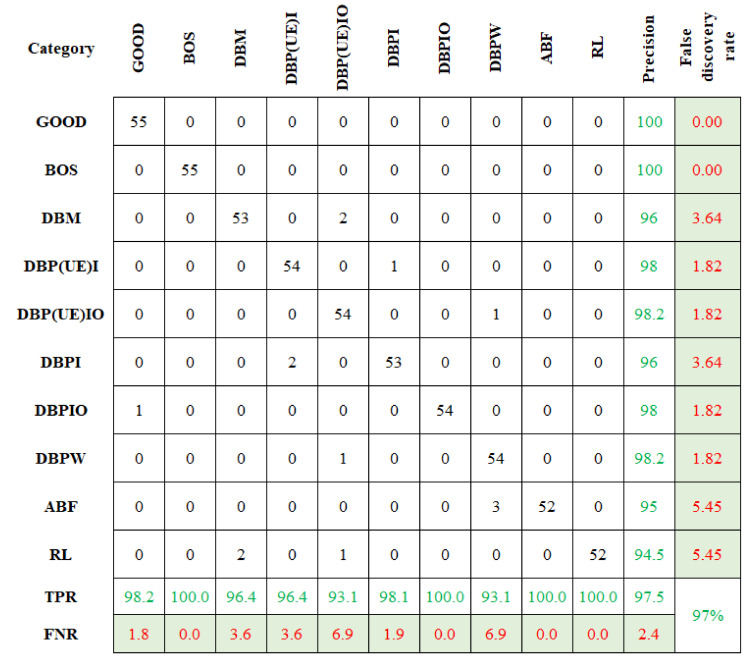
Confusion matrix for ND with histogram features.

**Figure 8 sensors-23-09093-f008:**
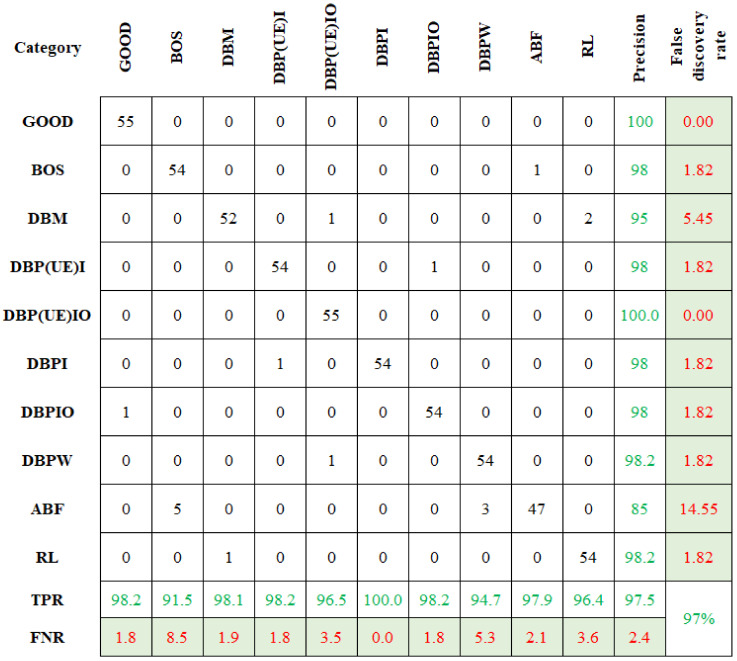
Confusion matrix for CBND with histogram features.

**Figure 9 sensors-23-09093-f009:**
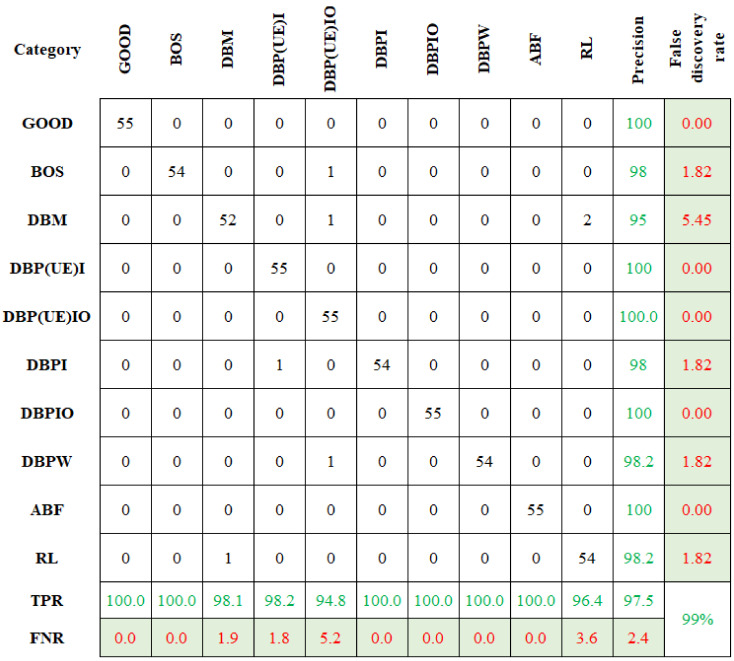
Confusion matrix for DNBND with histogram features.

**Figure 10 sensors-23-09093-f010:**
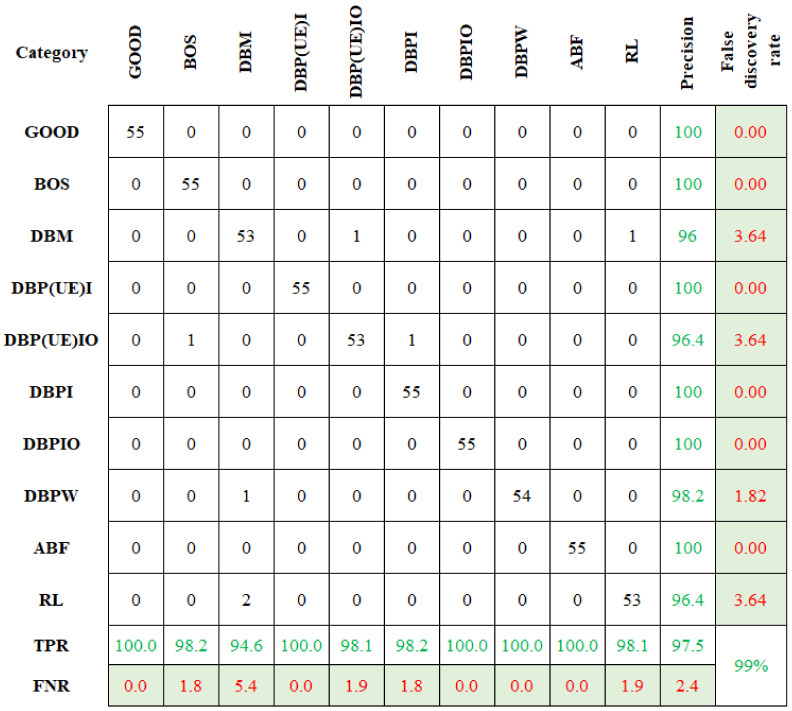
Confusion matrix for ND with Coiflets wavelet.

**Figure 11 sensors-23-09093-f011:**
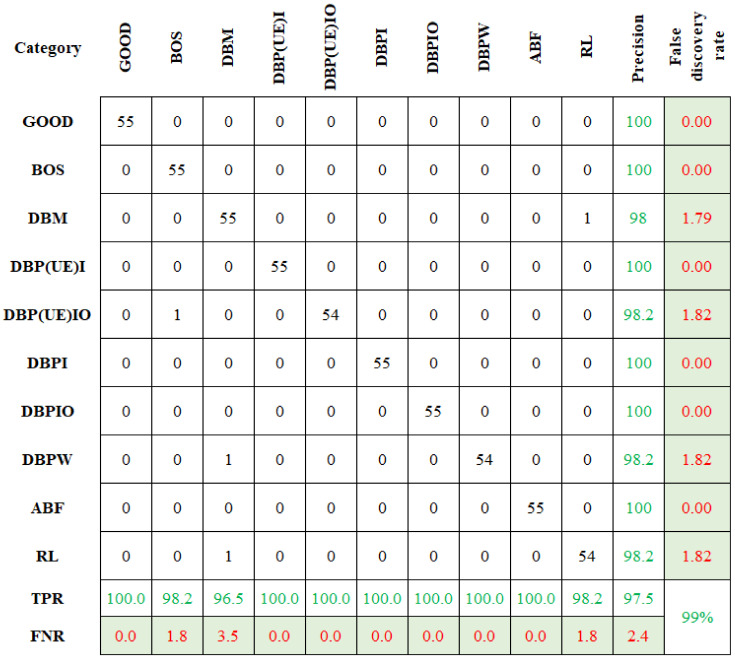
Confusion matrix for CBND with wavelet.

**Figure 12 sensors-23-09093-f012:**
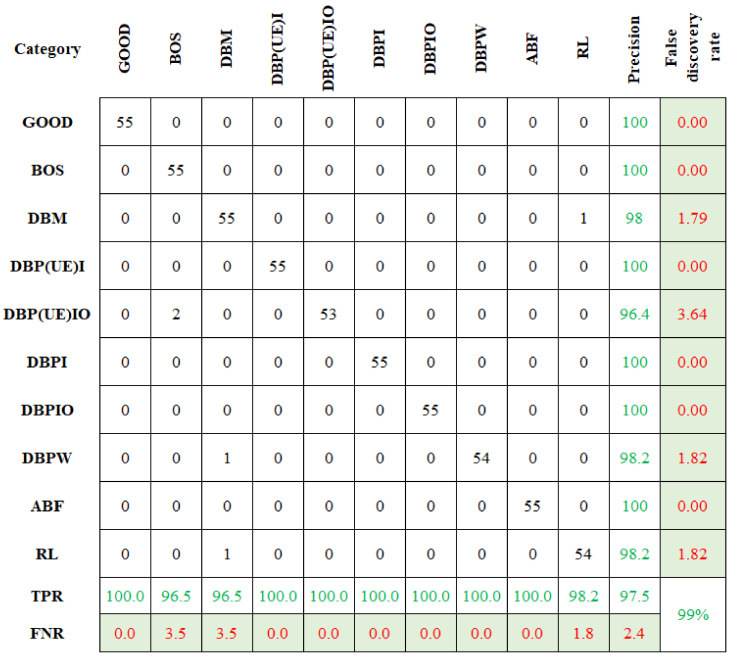
Confusion matrix for DNBND with Coiflets wavelet.

**Table 1 sensors-23-09093-t001:** Influence of no. of feature study with Coiflets wavelet.

No. of Features	Categorization Accuracy (%)		
	ND	CBND	DNBND
1	28.91	28	28.36
2	89.45	88.55	90
3	96	95.27	95.82
4	94.55	94.91	95.27
5	98	98.73	98.55
6	98	99.27	98.73
7	98.73	98.55	98.73
8	98.73	99.45	99.27
9	98.55	99.27	98.91

**Table 2 sensors-23-09093-t002:** Classification accuracy for various wavelet families.

Wavelet	Classification Accuracy (%)		
	ND	CBND	DNBND
Coif1L5	98.73	99.45	99.27
Dmey	99.27	99.09	99.09
Bior2.2L5	98.18	98.18	98.36
Db5L5	99.09	98.73	98.36
Haar	98.73	98.55	98.55
Sym3L5	98.73	98.55	98.55
Rbior2.8L5	86.73	85.82	86
Fk4L5	97.64	98.18	98.36

**Table 3 sensors-23-09093-t003:** Overall classification accuracy of ND family classifiers.

Classifier	Features	Total No. of Data	Correctly Categorized Instances		Incorrectly Categorized Instances		Kappa Statistic
ND	Histogram	550	540	98.18%	10	1.82%	0.9758
CBND	Histogram	550	542	98.55%	8	1.45%	0.9818
DNBND	Histogram	550	542	98.55%	8	1.45%	0.9818
ND	Coiflets wavelet	550	543	98.73%	7	1.27%	0.9859
CBND	Coiflets wavelet	550	547	99.45%	3	0.55%	0.9939
DNBND	Coiflets wavelet	550	546	99.27%	4	0.73%	0.9919

**Table 4 sensors-23-09093-t004:** Comprehensive accuracy by conditions—CBND with Coiflets wavelets.

Category	TPR	FPR	Precision	Recall	F-Measure	ROC Area
ABF	1	0	1	1	1	1
BOS	1	0.002	0.982	1	0.991	1
DBM	1	0.004	0.965	1	0.982	1
DBP(UE) inner	1	0	1	1	1	1
DB(UE) IO	0.982	0	1	0.982	0.991	1
DBPI	1	0	1	1	1	1
DBPIO	1	0	1	1	1	1
DBPW	0.982	0	1	0.982	0.991	1
GOOD	1	0	1	1	1	1
RL	0.982	0	1	0.982	0.991	1
Weighted Average	0.995	0.001	0.995	0.995	0.995	1

**Table 5 sensors-23-09093-t005:** Classification accuracy of ND classifiers.

S. No.	Name of the Classifier	Classification Accuracy (%)	
		Wavelet Features	Histogram Features
1	ND	99.20	97.50
2	CBND	99.45	97.00
3	DNBND	99.30	99.00

## Data Availability

The data used to support the findings of this study will be provided upon request and institutional approval.

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
