# Peer review of "A Comprehensive Approach for Detecting Brake Pad Defects Using Histogram and Wavelet Features with Nested Dichotomy Family Classifiers"

_sensors, 2023, doi:10.3390/s23229093_

Round 1

Reviewer 1 Report

Comments and Suggestions for Authors

Ensuring the reliability of hydraulic braking systems is crucial for the safe and efficient operation of commercial vehicles. This paper investigates the method of monitoring the vibration signal of the braking system using wavelet and histogram features. By analyzing and comparing the accuracy of Nested Dichotomy family classifiers in classifying brake system faults, it is recommended that CBND with wavelet features can be used for online hydraulic braking system fault diagnosis. The method studied in the paper is intuitive, concise and highly practical, with good application value.

Author Response

Sensors-2680492

Advanced Health Monitoring of Brake Systems: Leveraging Histogram and Wavelet Features in Vibration Signal Analysis with Nested Dichotomy Family Classifiers

Response to comments of Reviewer 1:

            We would like to thank the reviewer for the valuable and useful comments, which have helped us enhance our article's quality. We have revised the manuscript by addressing all of the suggestions and comments to the best of our ability. We are pleased to present our point-by-point responses to all of the comments below and briefly describe the revisions made.

General Comments:

Ensuring the reliability of hydraulic braking systems is crucial for commercial vehicles' safe and efficient operation. This paper investigates the method of monitoring the vibration signal of the braking system using wavelet and histogram features. By analyzing and comparing the accuracy of Nested Dichotomy family classifiers in classifying brake system faults, it is recommended that CBND with wavelet features can be used for online hydraulic braking system fault diagnosis. The method studied in the paper is intuitive, concise, and highly practical, with good application value.

Response:

We would like to thank the reviewer for their encouraging comments.

Reviewer 2 Report

Comments and Suggestions for Authors

Author Response

Sensors-2680492

Advanced Health Monitoring of Brake Systems: Leveraging Histogram and Wavelet Features in Vibration Signal Analysis with Nested Dichotomy Family Classifiers

Response to comments of Reviewer 1:

            We would like to thank the reviewer for the valuable and useful comments, which have helped us enhance our article’s quality. We have revised the manuscript by addressing all of the suggestions and comments to the best of our ability. We are pleased to present our point-by-point responses to the comments attached and briefly describe the revisions made.

Reviewer 3 Report

Comments and Suggestions for Authors

Not all Author affiliations are complete - no country

The description of the experimental research lacks information about the location of the accelerometer, its mounting method and the direction of measurement. It was also not explained how the pressure on the brake pedal was measured. Without this information, it is difficult to assess the reliability of the results obtained.

It is also not clear whether the given speed (about 30 km per hour) is the initial speed of the braking process or rather is it the maintaining speed during braking?

How was the wheel speed, which was 331 rpm, measured and controlled?

An example measurement result of the braking process is also missing, i.e. a time diagram of the vibration acceleration values.

I congratulate the authors on their ability to read the results from the graphs provided, but in my opinion the graphs in Figure 3 are largely identical. It is impossible to distinguish whether the result is for good technical condition "GOOD" or for damage: "BOS", "DBPI", "DBM".

Moreover, the drawings are numbered incorrectly!

The sample signal fragments shown in Figure 4 are too short. At the assumed wheel rotation speed of 331 rpm, the wheel makes approximately 5.5 revolutions per second. At 25 kHz sampling, the 2041 samples shown represent approximately 0.08 seconds. So it's not even half a revolution of the wheel! There is also no information about the synchronization of measurements, which means that each time the result may be for a different part of the brake disc!

Due to the above comments, I believe that the errors made during the experiment disqualify further results of their analyses.

The article is not suitable for printing.

Author Response

Sensors-2680492

Advanced Health Monitoring of Brake Systems: Leveraging Histogram and Wavelet Features in Vibration Signal Analysis with Nested Dichotomy Family Classifiers

            We would like to thank the reviewer for the valuable and useful comments, which have helped us enhance our article's quality. We have revised the manuscript by addressing all of the suggestions and comments to the best of our ability. We are pleased to present our point-by-point responses to the comments attached and briefly describe the revisions made.

Round 2

Reviewer 2 Report

Comments and Suggestions for Authors

1- The authors mentioned that they considered the third comment and corrected the paper. but in the revised version, it is not corrected yet.

Author Response

Response to comments of Reviewer 1:

            We would like to thank the reviewer for the valuable and useful comments, which have certainly helped us to enhance the quality of our article. We have revised the manuscript by addressing all of the suggestions and comments to best our ability. We are pleased to present our point-by-point responses to all of the comments below together with brief description of the revisions made.

Comment #1: The authors mentioned that they considered the third comment and corrected the paper. but in the revised version, it is not corrected yet.

Response #1

We agree with the reviewer’s suggestion. We acknowledge that there seems to be an oversight in the revised version of the paper. Modified sentence is added in the revised manuscript.

Details added in the revised manuscript in line no.165 page # 4:

The wheel speed, which was 331 rpm, was measured and controlled using a tachometer. The brakes were applied gradually and evenly to prevent RPM fluctuations, and, importantly, as the brakes were applied, minor manual adjustments were made to the throttle to ensure a consistent RPM was maintained”.

Reviewer 3 Report

Comments and Suggestions for Authors

After the authors introduced corrections in the content of the article, it is correct.

But there are still some necessary additions and changes. The content lacks a clear indication that the method only applies to the detection of brake pad (or brake caliper) wear. The influence of the brake disc (its technical condition) is not fully taken into account. Many different types of brake disc damage (warping, uneven wear, corrosion, cracks) strongly influence the vibrations measured at the brake caliper.

Therefore, I propose to change the title and purpose of the work. The described method cannot be the basis for a comprehensive diagnosis of the braking system as indicated in the title.

Author Response

Response to comments of Reviewer 2:

General Comments:

After the authors introduced corrections in the content of the article, it is correct.

Response:

We would like to thank the reviewer for his/her encouraging comments.

Comment #1:

But there are still some necessary additions and changes. The content lacks a clear indication that the method only applies to the detection of brake pad (or brake caliper) wear. The influence of the brake disc (its technical condition) is not fully taken into account. Many different types of brake disc damage (warping, uneven wear, corrosion, cracks) strongly influence the vibrations measured at the brake caliper.

Therefore, I propose to change the title and purpose of the work. The described method cannot be the basis for a comprehensive diagnosis of the braking system as indicated in the title.

Response #1

We appreciate the reviewer's feedback and have taken it into consideration. Our primary focus is on detecting brake pad defects, and we have acknowledged the reviewer's valid point regarding the influence of brake disc defects on the brake caliper's vibration. As such, we are revising the title and purpose of our work to better reflect our scope: "A Comprehensive Approach for Detecting Brake Pad Defects Using Histogram and Wavelet Features with Nested Dichotomy Family Classifiers." This change clarifies that our method pertains exclusively to brake pad defects, addressing the reviewer's concerns and providing a more accurate representation of our research focus.